# Vaccines for Human Schistosomiasis: Recent Progress, New Developments and Future Prospects

**DOI:** 10.3390/ijms23042255

**Published:** 2022-02-18

**Authors:** Adebayo J. Molehin, Donald P. McManus, Hong You

**Affiliations:** 1Department of Microbiology and Immunology, College of Graduate Studies, Midwestern University, Glendale, AZ 85308, USA; amoleh@midwestern.edu; 2Department of Immunology, QIMR Berghofer Medical Research Institute, Brisbane, QLD 4006, Australia; don.mcmanus@qimrberghofer.edu.au

**Keywords:** schistosomiasis, clinical vaccine development, mRNA vaccine

## Abstract

Schistosomiasis, caused by human trematode blood flukes (schistosomes), remains one of the most prevalent and serious of the neglected tropical parasitic diseases. Currently, treatment of schistosomiasis relies solely on a single drug, the anthelmintic praziquantel, and with increased usage in mass drug administration control programs for the disease, the specter of drug resistance developing is a constant threat. Vaccination is recognized as one of the most sustainable options for the control of any pathogen, but despite the discovery and reporting of numerous potentially promising schistosome vaccine antigens, to date, no schistosomiasis vaccine for human or animal deployment is available. This is despite the fact that *Science* ranked such an intervention as one of the top 10 vaccines that need to be urgently developed to improve public health globally. This review summarizes current progress of schistosomiasis vaccines under clinical development and advocates the urgent need for the establishment of a revolutionary and effective anti-schistosome vaccine pipeline utilizing cutting-edge technologies (including developing mRNA vaccines and exploiting CRISPR-based technologies) to provide novel insight into future vaccine discovery, design, manufacture and deployment.

## 1. Introduction

Parasitic diseases remain a major cause of morbidity and mortality globally, disproportionately affecting people living in the poorest regions of the world. Significant progress has been made in reducing the burden of human parasitic infections through the Millennium Development Goals and the Sustainable Development Goals [1]. However, changing environments and population dynamics pose new challenges, and this is particularly true of schistosomiasis, a neglected tropical disease caused by parasitic flatworms of the genus *Schistosoma*. Six geographically distinct species of *Schistosoma* are responsible for infections in humans, resulting in significant morbidity and contributing to over 290,000 deaths per year [2]. The symptoms of schistosomiasis are chronic, insidious and menacing due to prolonged egg deposition and consequent inflammation and granulomatous reactions in affected tissues such as the liver and intestine. An estimated 3.3 million disability-adjusted life years (DALYs) have been attributed to schistosomiasis, although some estimates are much higher, even reaching DALYs that may exceed 70 million [2,3,4,5]. In addition to the public health burden, the disease also imposes a heavy socio-economic cost on affected communities [6]. Currently, schistosomiasis is endemic in 78 countries, with over 250 million people living with the disease and an estimated 800 million people at risk of being infected [7].

To date, schistosomiasis control efforts have centered on strategies ranging from disease treatment to managing complications and limiting disease spread through various public health efforts such as health education, snail intermediate host control, and water, sanitation and hygiene (WASH) programs [8,9,10]. The effect of the targets set by the World Health Organization for global schistosomiasis control, based on large-scale mass drug administration (MDA) of praziquantel (PZQ), has been suboptimal due to a myriad of logistical challenges, including a shortfall in drug delivery and adherence, sustained reinfection rates and inadequate infrastructure [11,12,13,14]. While these MDA programs have been somewhat effective in the short term, they are, however, unsustainable in the long term. Additionally, interruptions in these MDA programs often lead to the occurrence of severe rebound disease, particularly in high-transmission areas [12,15]. PZQ helps to control morbidity in patients receiving treatment by killing established adult schistosomes; however, because it is ineffective against juvenile worms and does not prevent reinfection, the overall effect on disease transmission is transient, as prevalence returns to baseline levels within a very short period of time [2]. Despite the widespread use of PZQ over the past 40 years, the number of people infected, particularly in Africa, has not decreased substantially [16]. Reports of schistosomiasis transmission in certain previously schistosomiasis-free areas of Europe [17,18,19], in addition to the shortfalls of current control measures, have only heightened the urgency of reevaluating current control approaches if meaningful progress is to be made towards achieving the Millennium Development Goals of schistosomiasis elimination.

Historically, vaccine administration has been the most cost-effective way of preventing human infections with various pathogens in the long term. In fact, the impact of vaccination on global health has been highly significant, on par with the introduction of clean water and proper sanitation [20,21]. In order to achieve sustainable schistosomiasis control targets, it is clear that an integrated, multifaceted approach will be required, with an effective vaccine serving as a major fulcrum [21,22,23,24]. Several hurdles remain, as we do not yet have a licensed product for human use. There is, however, reason for cautious optimism provided by some of the encouraging vaccine efficacy data obtained from experimental and human challenge models of schistosomiasis [21,25].

Recently published reviews have covered key aspects of schistosome immunopathobiology, host–schistosome interactions and aspects of disease management [2,25,26]. Here, we discuss recent progress in clinical schistosomiasis vaccine development and provide an update on new technologies employed in schistosomiasis vaccine discovery.

## 2. Schistosomiasis Vaccines: Update on Clinical Development

Proposals for the Preferred Product Characteristics for a prophylactic schistosomiasis vaccine suggest that an effective vaccine should reduce morbidity and disease transmission by at least 75% [22,27]. It is important to note that the goal is not to achieve sterile immunity, as schistosomes do not replicate in their mammalian hosts; therefore, a vaccine with even partial protective efficacy would significantly reduce disease burden and subsequent transmission. A candidate vaccine that preferentially kills egg-producing female worms while preserving natural immunity induced by nonpathogenic male worms would be an added advantage [27,28]. To date, several hundred candidate antigens have been identified and tested against one or more of the three major clinically relevant *Schistosoma* species (*S. mansoni*, *S. haematobium* and *S. japonicum*) in murine and/or nonhuman primate models of infection. Suffice it to say that many of these candidate antigens have not made it beyond the preclinical stage of development partly due to the fact that many of these antigens were evaluated in animal models with documented inherent flaws [29] and with adjuvants being chosen for the vaccine formulations [22,30]. There is now a strong case being made for promising vaccine candidates identified from murine models to be validated in nonhuman primates before embarking on clinical development, because these animals adequately reflect the immunopathogenesis observed in humans [21,29,31,32]. However, conducting efficacy studies with nonhuman primates is a huge undertaking with the high cost and ethical justification for the use of these animals being major limitations. Furthermore, consideration of the complex immunological interactions between vaccine, co-infections, prior schistosome exposure and post-PZQ treatment in endemic populations is critical in clinical trial design and application policies [33]. In addition, host IgE production, which is associated with the risk of allergic reactions and the potential of aggravating granulomas and fibrosis by egg-induced responses [34], makes the development of an anti-schistosomiasis vaccine more challenging. Despite the seemingly insurmountable obstacles that have in the past and still beset schistosomiasis vaccine development, there is cause for optimism. Firstly, there are four candidate antigens (discussed below) currently in different phases of clinical development: *Schistosoma haematobium* 28-kD glutathione S-transferase (Sh28GST/Alhydrogel)) [35,36], *Schistosoma mansoni* 14-kDa fatty acid-binding protein (Sm14/GLA-SE) [37], *Schistosoma mansoni* tetraspanin (Sm-TSP-2/Alhydrogel) [38] and *Schistosoma mansoni* calpain (Sm-p80/GLA-SE). Secondly, progress in adjuvant technologies has also shown some promise due to the availability of novel adjuvants that are capable of selectively activating certain aspects of the host immune system that are critical to long-term vaccine-mediated immunity [39]. In addition, the recently established *Schistosoma mansoni* controlled human infection model [40,41] will undoubtedly accelerate the process of vaccine development and also provide an invaluable platform for the identification of novel vaccine candidates.

### 2.1. Schistosoma Mansoni Tetraspanin (Sm-TSP-2)

Tetraspanins (TSPs) are surface membrane and scaffolding proteins in schistosomes, and they are involved in the regulating funcitons of other membrane proteins, in the trafficking and tegument formation [42,43]. There are two main tetraspanin types found in *S. mansoni*, Sm-TSP-1 and Sm-TSP-2 [44]. Structurally, the schistosome tetraspanins are composed of four transmembrane domains connected by extracellular loops that are readily accessible to the host immune system [44]. Based on the fact that TSP-2 (and not TSP-1) is strongly recognized by IgG1 and IgG3 from putative resistant individuals and not by infection-naïve or chronically infected individuals, preclinical studies focused on the development of Sm-TSP-2 [44]. Efficacy studies in mice showed that immunization with rSm-TSP-2 resulted in a significant reduction in adult *S. mansoni* worm (57%) and liver egg burdens (64%). Other studies using either Sm-TSP-2 or a chimera of Sm-TSP-2 and 5B (the immunogenic region of hookworm aspartic protease vaccine antigen, *Na*-APR-1) formulated with alum/CpG induced significant levels of protection against *S. mansoni* infections, with a 25–58% and 27–56% reduction in worm and tissue egg burdens, respectively; these vaccines were also associated with the induction of vaccine-mediated humoral immune responses [45,46]. A similar study using a Sm-TSP-2/Sm29 chimera also resulted in enhanced protection in immunized animals with concomitant production of Th1-type immune responses associated with the protection observed [46]. Importantly, Sm-TSP-2-specific (and Sm-TSP-2/5B chimera) IgE antibodies were undetectable in sera from chronically infected people living in areas of *S. mansoni*/hookworm co-endemicity [46].

An initial Phase 1a dose-escalation study was conducted to assess the safety, reactogenicity and immunogenicity of Sm-TSP-2 formulated on aluminum hydroxide adjuvant (Alhydrogel^®^ Frederikssund, Denmark) with or without glucopyranosyl lipid adjuvant in an aqueous formulation (GLA-AF) in healthy adults from a *S. mansoni* nonendemic area [38]. Results from the study showed that the vaccine was safe and well-tolerated with no vaccine-related adverse events. The vaccine induced a dose-dependent Sm-TSP-2-specific IgG peaking after the second booster. A subsequent dose-escalation Phase 1b study was carried out to assess the safety, immunogenicity and tolerability of SmTSP-2/Alhydrogel^®^ with or without AP 10-701 in healthy adults exposed to *S. mansoni* natural infections, but the results are yet to be published (https://clinicaltrials.gov/ct2/show/NCT03110757, 12 April 2017). Safety, immunogenicity and efficacy testing of SmTSP-2/Alhydrogel^®^ with or without AP 10-701 in healthy Ugandan adults is currently under investigation in Phase 1 and 2b trials, which are expected to be completed by early 2025 (https://clinicaltrials.gov/ct2/show/NCT03910972, 10 April 2019).

### 2.2. Schistosoma Mansoni Calpain (Sm-p80)

Calpain, a cysteine protease consisting of a catalytic (large) subunit and a proteolytic (small) subunit [47], is highly expressed in the tegument of adult schistosomes and in other lifecycle stages [48]. Calpain plays a major role in host immune evasion by schistosomes through its involvement in tegument biogenesis and renewal [49]. Due to its accessibility to the host immune system and the critical role it plays in the survival of schistosomes within the host, the large subunit of calpain, Sm-p80, was identified as a candidate vaccine and subjected to preclinical development. Several vaccine efficacy studies using the Sm-p80 antigen in various vaccine/adjuvant formulations and strategies in murine and nonhuman primate models of *S. mansoni* infection and disease showed that the Sm-p80-based vaccine offered significant prophylactic, therapeutic, anti-pathology, cross-species and transmission-blocking protection in vaccinated animals [28,32,48,50,51,52,53]. A recent preclinical trial in specific pathogen-free baboons revealed an Sm-p80-mediated preferential killing of adult female worms (93%) resulting in a 90% decrease in overall tissue egg load in immunized animals; the authors also reported a significant vaccine-mediated reduction in fecal egg excretion [54]. The efficacy of the Sm-p80 vaccine was also evaluated in a scenario of chronic schistosomiasis, PZQ treatment and re-exposure to *S. mansoni* infection. Sm-p80-immunized baboons had a significant reduction of 38%, 72% and 49% in liverand small and large intestinal egg burdens, respectively, with corresponding reductions in egg viability of 60%, 49% and 82%. Importantly, Sm-p80-specific IgE antibodies are not detectable in sera from individuals living in *S. mansoni* endemic areas [55,56], thereby removing the potential risk of vaccine-induced hypersensitivity reactions. A phase 1a clinical trial using Sm-p80 formulated in GLA-SE (SchistoShield^®^, Seattle, DC, USA) has commenced in infection-naïve adults in the United States, and this will be followed by a Phase 1b dose-escalation trial among African adults, with a planned future age de-escalation study in school-aged children [57].

### 2.3. Schistosoma Mansoni 14-kDa Fatty Acid-Binding Protein (Sm14)

Fatty acid-binding proteins (FABPs), ubiquitously expressed by all lifecycle stages of schistosomes, allow for the acquisition of host-derived essential fatty acids and sterols, since blood flukes lack oxygen-dependent pathways [58]. Individuals that are naturally resistant to schistosome infections demonstrate a robust Th1 immune response to the *S. mansoni* 14-kDa fatty acid-binding protein (Sm14) with the increased Th1-type immunity profile correlating with decreased liver pathology [59,60]. Furthermore, Sm14-immunized outbred Swiss mice and New Zealand white rabbits exhibited a 67–93% reduction in worm burden following *S. mansoni* cercarial challenge [61]. Following these preclinical studies, recombinant Sm14, formulated in glucopyranosyl lipid adjuvant-stable emulsion (GLA-SE) (Sm14/GLA-SE,), was tested in a Phase 1 clinical trial whereby its safety and immunogenicity were assessed in healthy subjects from a nonendemic area of Brazil [37]. Overall, the vaccine was highly immunogenic and well-tolerated with few mild adverse events and no detectable vaccine-induced IgE antibodies. A follow-up phase 2a study among adults living in a schistosome-endemic region of Senegal showed that Sm14/GLA-SE was safe and resulted in 92% seroconversion after the third immunization (https://clinicaltrials.gov/ct2/show/NCT03041766, 3 February 2017). Based on the results of the Phase 2a trial, a phase 2b study in school-aged children living in the same endemic area of Senegal was conducted and completed in 2019, but the results are yet to be released (https://clinicaltrials.gov/ct2/show/study/NCT03799510, 10 January 2019).

### 2.4. Schistosoma Haematobium 28-kDa Glutathione S-Transferases (Sh28GST)

Glutathione S-transferases (GST) are enzymes involved in many processes associated with metabolic and detoxification pathways [62]. In schistosomes, these enzymes (28GSTs) play critical roles in host immune modulation during infection, including annulling the capacity of epidermal Langerhans cells to move to the draining lymph nodes [63], the binding of testosterone and the detoxification of xenobiotic compounds [64,65]. Since its characterization in the late 1990s, data from several efficacy studies using the Sh28GST vaccine in murine and nonhuman primate models of *S. haematobium* infection and disease showed protective immunity following cercarial challenge with profound effects on tissue egg pathology and excretion [35,57]. An initial Phase 1 study assessed the safety, tolerability and immunogenicity of recombinant Sh28GST (rSh28GST) adsorbed to Alhydrogel (Bilhrvax^®^ Lille, France) in healthy Caucasian adults, and showed that the vaccine was well-tolerated and elicited a strong Th2-biased immune response [66]. A subsequent Phase 2 trial that assessed the co-administration of Bilharvax^®^ and praziquantel (PZQ) in *S. haematobium*-infected adults and children also revealed that the vaccine was safe [66]. These findings precipitated a Phase 3 trial in which the safety, immunogenicity and protective efficacy of Bilharvax^®^ was evaluated in PZQ-treated infected Senegalese school-aged children. Unfortunately, the authors reported suboptimal efficacy levels despite high levels of seroconversion in Bilharvax^®^-immunized individuals [36]. The authors suggested that the lack of efficacy observed may have been due partly to repeated PZQ treatment interference and/or the vaccine-administration regimen used, which favored the blocking of IgG4 production rather than the induction of protective IgG3 antibodies [36]. Planned trials in the future should consider assessing Bilharvax^®^ without PZQ and perhaps utilizing another Th1-biased adjuvant.

## 3. Challenges in the Development of Schistosomiasis Vaccines

Parasitic diseases (including schistosomiasis) mainly affect the poorest regions of the world with low-base economies, with most parasites causing chronic illnesses and disabilities that generally do not directly kill their hosts, with one notable exception being malaria. As a consequence, there has been relatively limited interest in advancing novel platforms for schistosomiasis vaccine development. Furthermore, the acquisition of effective anti-schistosomiasis vaccines has proven to be extremely challenging given the fact that traditional vaccine platforms are ill-suited due to the complexity of the schistosome life cycle, the parasite’s ability to evade the host immune system and the fact that animal models do not adequately represent accurate protective immune responses compared with those generated in natural mammalian hosts. Indeed, most of the current understanding of schistosomiasis immunology has been established through studies conducted in mice, which likely do not provide an accurate representation of the responses generated in natural, outbred mammalian hosts in endemic regions [67]. The same arguments apply to vaccine trials involving challenge infections with schistosome cercariae, which are difficult to kill via acquired protective immune responses in the mouse model [29]. Additionally, a number of key biological differences exist between mice (permissive hosts) and natural schistosome hosts [68]. In the translation of promising vaccine candidates, first identified using murine models, it is critical to evaluate their protective efficacy in nonhuman primates (for *S. mansoni* and *S. haematobium*) or natural hosts (e.g., bovines for *S. japonicum*) [21,29,31,32], although this is a more complicated undertaking with high associated costs leading to major limitations in the development of clinical vaccines against schistosomiasis.

In addition, it is important to continue identifying new target schistosome antigens by exploiting new, cutting-edge technologies. Different approaches (shown in Table 1), including transcriptomics and DNA microarray profiling [69,70,71,72], proteomics [73,74,75,76,77], immunomics [78,79,80,81], glycomics [82,83], exosomics [84,85,86] and gene suppression [70,87,88,89,90], have been used in the past decade to identify novel vaccine targets. Complete genomic sequences are available for *S. japonicum* [90], *S. mansoni* [72] and *S. haematobium* [71], but of the approximately 13,000 protein-encoding genes present in each species, very few have been functionally characterized. A major challenge for researchers in mining genomes is the lack of suitable tools to effectively characterize schistosome gene products as potential vaccine targets and to translate them into urgently needed interventions. Recently, using a large-scale RNA interference (RNAi) screening system, Wang et al. [91] examined the functions of 2216 genes in adult *S. mansoni* and identified 261 genes with phenotypes affecting neuromuscular function, tissue integrity, stem cell maintenance and parasite survival. The group discovered two important protein kinases (TAO and STK25) that play key roles in maintaining muscle-specific messenger RNA transcription [91]. Loss of activity of either of these kinases would result in muscular function defects leading to paralysis and worm death in the mammalian host. This novel approach expedites therapeutic development by uncovering new phenotypes of novel targeted genes. New technologies, such as the rapid development of the clustered regularly interspaced short palindromic repeats- (CRISPR)/CRISPR-associated protein 9 (Cas9)) mediated editing system, provides a powerful genetic approach for interrogating genomes and defining the function of key genes in various organisms by triggering specific and heritable genome editing [92]. For developing this technology, Emmanuelle Charpentier and Jennifer A. Doudna were awarded the Nobel Prize in Chemistry in 2020.

The CRISPR/Cas9 editing system has been successfully established in *S. mansoni* by targeting three different genes, including gene omega-1 [93], a secreted T2 ribonuclease crucial for Th2 polarization and granuloma formation; acetylcholinesterase [94], a key enzyme and the target of a number of currently approved and marketed anthelminthic drugs; and the *SULT-OR* gene [95], in which mutations confer resistance to the anti-schistosome drug oxamniquine. These studies have been well-received, emphasizing the value of CRISPR/Cas9 editing as a powerful tool to precisely target and deactivate genetic information in schistosomes. Aiming to improve the modification efficiency of CRISPR/Cas9 editing in schistosomes is critical, given the structural complexities that exist within the different life cycle stages of the multiple-celled parasite [96], a common feature for helminth parasites. This pivotal technology would undoubtedly provide the blueprint for programmed gene editing and functional genomics studies in schistosomes while serving as a vehicle to identify novel anti-schistosome vaccine candidates.

**Table 1 ijms-23-02255-t001:** Summary of the technologies that have been used in the development of anti-schistosomiasis vaccines.

Technologies Applied in the Identification of Vaccine Targets	Examples of Procedures Utilised	References
Transcriptomics and DNA microarray profiling	RNA-sequencing, next generation sequencing	[69,70,71,72]
Proteomics	Chromatography-based methods, antibody-based methods	[73,74,75,76,77]
Immunomics	ELISPOT, Immunomic microarrays, T- and- B-cell-epitope mapping tools	[78,79,80,81]
Glycomics		[82,83]
Exosomics		[84,85,86]
Gene suppression	RNA interference, vector-based silencing, lentiviral transduction	[70,87,88,89,90]
Gene editing	CRISPR/Cas9	[93,94,95]
**Technologies used in vaccine delivery**		
Recombinant protein vaccines or bivalent vaccines	Smp80, Sm14, Sm-TSP-2, Sm14/Sm29, Sm14/Sm-TPS-2/Sm29/Smp80	[44,54,61,97,98,99]
Synthetic multi-epitope peptides	Sm14	[59,60,100]
DNA-based vaccines	SjCTPI, Smp80	[101,102,103]
Irradiated cercarial vaccines		[104,105,106]
New adjuvants	R848, TLR7/8 agonist, CpG-ODN, QuilA, GLA-SE, alum, poly(I:C)	[30,38,44,107,108,109]

Note: ELISPOT, enzyme-linked immune absorbent spot; SjCTPI, *S. japonicum* triose-phosphate isomerase.

## 4. mRNA Vaccine Technology: Forging New Frontiers in Vaccine Capabilities

The unparalleled rapid and successful development of highly effective SARS-CoV-2 mRNA vaccines has shown that mRNA vaccines are safe and highly potent in evoking a strong, effective and well-defined immune response [110]. In stark contrast to the costly, long processing time of conventional vaccines (as listed in Table 1) and the considerable difficulty inherent in the purification of recombinant proteins, in vitro transcribed mRNA vaccines can be made quickly and easily, allowing for multivalent combinations to enable synergistic effects that further enhance immunity [111]. The mRNA platform offers enhanced stability and targeted antigen expression and has already proven successful against challenging diseases where conventional technology has failed. To date, mRNA vaccines against three different single-celled parasites (*Plasmodium* malaria [112,113], *Leishmania donovani* [114] and *Toxoplasma gondii* [115]) have been developed. Anti-*Plasmodium* mRNA vaccines targeting the circumsporozoite protein (PfCSP) and cytokine macrophage migration inhibitory factor (PMIF) have been successfully tested; they induce strong specific CD4+ T cell responses and high titer IgG antibodies, resulting in the generation of protective immunity against malaria infection in mice [112,113]. Heterologous mRNA has also been used to vaccinate mice against *L. donovani* infection, resulting in a significant reduction in liver parasite burden through inducing strong IFN-γ secretion and antigen-specific Th1 responses by splenocytes [114]. A self-amplifying mRNA-LNPs approach was also utilized to develop an effective vaccine against *T. gondii* infection in mice [115]. Immunization of animals with mRNA vaccines can promote the secretion of type I interferons that creates a milieu that favors the Th1 response over Th2 [116], which has been observed in a number of schistosomiasis vaccine research reports [46,117], indicating a highly specific IFN-γ (Th1) response correlating with a high level of protection against schistosome infection.

Encouragingly, a multivalent mRNA vaccine encoding 19 salivary proteins (19ISP) from *Ixodes scapularis* black-legged ticks (which can transmit many pathogens that cause human disease, including the Lyme disease agent *Borrelia burgdorferi*) has been recently developed [118]. Guinea pigs immunized with lipid nanoparticle-containing nucleoside-modified mRNAs encoding 19ISP elicited a strong specific antibody response and induced protection against tick challenge through the provision of robust tick immunity, which included early erythema after tick placement on the animals and rapid tick detachment, along with severely impaired tick feeding and low engorgement weights [118]. This study provides solid evidence to show that the goal of developing a multivalent mRNA vaccine targeting multiple genes against multicellular organisms (including the schistosomes) is achievable. It provides critical insight into the various factors affecting the protective efficacy elicited by mRNA vaccines, features that are likely to be required to develop similar vaccines against schistosomiasis and other helminth diseases.

## 5. Conclusions

Schistosomiasis remains a poverty-promoting and stigmatizing condition occurring mainly in rural areas of low-income countries. To improve on current control measures for schistosomiasis and to lead to its eventual elimination, an effective vaccine together with the deployment of other interventions, including chemotherapy, improved water, sanitation and hygiene, snail control, better health education and accurate diagnostics, will be required [119]. Although a large number of vaccine candidates have been identified, very few have made it to clinical trials, and these may not provide the level of protective immunity required. A revolutionary and effective anti-schistosome vaccine pipeline is urgently needed to develop and test novel vaccine antigens at high-throughput involving the identification and targeting of appropriate vaccine candidates through cutting-edge technologies, taking advantage of the mRNA vaccine platform that has been enlisted to generate the highly effective COVID mRNA vaccines and the use of suitable animal models for immunological analysis and the determination of vaccine effectiveness.

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
