# Peer review of "Vaccines for Human Schistosomiasis: Recent Progress, New Developments and Future Prospects"

_ijms, 2022, doi:10.3390/ijms23042255_

Round 1

Reviewer 1 Report

Estimated Authors of the paper "update on clinical vaccines for schistosomiasis and the potential of mRNA vaccine technology to defeat the disease",

I've read your contribution with high interest. In this narrative review, Molehin, McManus and You have summarized available evidence on the current state-of-art about vaccines against Schistosoma, have explained the main pro and cons of the various potential targets, and shed some insights about the potential use of new technologies, including mRNA based vaccines.

The review is well written, in plain English, and the references are both consistent with the main topic (particularly significant because of the non-systematic design of this paper) and up-to-date. 

In fact, from my point of view as a reviewer for IJMS, I've no specific requirements about potential improvements of the main text.

However, I've some concerns regarding the main title of this review.

In fact, Authors should modify the main title in order to clarify the narrative nature of this review; moreover, the inclusion of mRNA vaccines in the main title may be quite misleading, as the topic (albeit properly addressed) is only  discussed in the last section of the study, after an extensive and well designed discussion on other issues on vaccines for Schistosoma. In other words, I would recommend to either drop the reference to mRNA vaccines in the main title, or modify as follows: "Update on clinical vaccines for schistosomiasis and the potential of mRNA vaccine technology to defeat the disease" --> "Vaccines for human schistosomiasis: a narrative review on new developments and recent progresses" or similar.

Author Response

Response: Thank you very much for your comments.
We have modified the title to: Vaccines for human schistosomiasis: recent progress, new developments and future prospects.

Reviewer 2 Report

The manuscript brings to the attention schistosomiasis, a disease caused by human trematode blood flukes that represent serious tropical parasitic diseases that rely only on the treatment with praziquantel and prevention. The manuscript aims to provide a comprehensive review of the literature for the progress made up to date regarding a new vaccine for this disease. The manuscript is well written and provides very useful information for the research community. Some suggestions to improve the manuscript.

  1. A table that summarized all the technology that was tested for the development of the schistosomiasis vaccine should be added.
  2. The schematic representation of the mechanism of toxicity of parasites in the human body has to be included.

Author Response

  • A table that summarized all the technology that was tested for the development of the schistosomiasis vaccine should be added.

Response: As suggested, we have added Table 1 to summarise the technologies used for development of anti-schistosomiasis vaccines as discussed in the Manuscript.

  • The schematic representation of the mechanism of toxicity of parasites in the human body has to be included.

Response: Thank you for the comment. The main focus of this review is on clinical schistosomiasis vaccine development and emerging technologies. The suggested point of “a schematic representation of the mechanism of toxicity of parasites in the human body” has been discussed in depth in the published review articles: McManus et al, 2018 (Nat Rev. Dis. Primers) and McManus et al, 2020 (Semin. Immunopathol ) which were cited in the original MS as reference 2 and reference 25, respectively. To draw prospective reader’s further attention we have added sentences (line 100-105) to briefly address this issue as required by Reviewer 2.

Round 2

Reviewer 2 Report

The authors addressed my comments. The manuscript is improved and ready for acceptance.